# Promoting Vaccination in India through Videos: The Role of Humor, Collectivistic Appeal and Gender

**DOI:** 10.3390/vaccines10071110

**Published:** 2022-07-12

**Authors:** Amelia M. Jamison, Rajiv N. Rimal, Rohini Ganjoo, Julia Burleson, Neil Alperstein, Ananya Bhaktaram, Paola Pascual-Ferra, Satyanarayan Mohanty, Manoj Parida, Sidharth Rath, Eleanor Kluegel, Peter Z. Orton, Daniel J. Barnett

**Affiliations:** 1Department of Health, Behavior and Society, Bloomberg School of Public Health, Johns Hopkins University, Baltimore, MD 21205, USA; rimal@jhu.edu (R.N.R.); jburles4@jhu.edu (J.B.); abhakta2@jhu.edu (A.B.); dbarnet4@jhu.edu (D.J.B.); 2Department of Biomedical Laboratory Sciences, George Washington University, Ashburn, VA 20147, USA; rganjoo@egwu.edu; 3Department of Communication, Loyola University of Maryland, Baltimore, MD 21210, USA; nalperstein@loyola.edu (N.A.); ppascualferra@loyola.edu (P.P.-F.); emkluegel@loyola.edu (E.K.); 4Satya Nagar, Bhubaneswar 751007, Odisha, India; satya.dcor@gmail.com (S.M.); manoj.dcor@gmail.com (M.P.); 5Swasthya Plus Network, Chandrasekharpur, Bhubaneswar 751017, Odisha, India; sidharth@odicast.com; 6Wellflix Inc., Hillsborough, NC 27278, USA; orton@wellflix.net; 7Department of Environmental Health & Engineering, Bloomberg School of Public Health, Johns Hopkins University, Baltimore, MD 21205, USA

**Keywords:** vaccine hesitancy, vaccine confidence, collectivism, humor, message testing

## Abstract

Vaccination hesitancy is a barrier to India’s efforts to control the COVID-19 pandemic. Considerable resources have been spent to promote COVID-19 vaccination, but evaluations of such efforts are sparse. Our objective was to determine how vaccine videos that manipulate message appeal (collectivistic versus individualistic), tone (humorous versus serious), and source (male versus female protagonist) toward vaccines and vaccination. We developed eight videos that manipulated the type of appeal (collectivistic or individualistic), tone of the message (humor or serious), and gender of the vaccine promoter (male or female) in a 2 × 2 × 2 between-subjects experiment. Participants (*N* = 2349) were randomly assigned to watch one of eight videos in an online experiment. Beliefs about vaccines and those about vaccination were obtained before and after viewing the video. Manipulation checks demonstrated that each of the three independent variables was manipulated successfully. After exposure to the video, beliefs about vaccines became more negative, while beliefs about vaccination became more positive. Humor reduced negative beliefs about vaccines. Collectivism and protagonist gender did not affect beliefs about vaccines or vaccination. Those able to remember the protagonist’s gender (a measure of attention) were likely to develop favorable beliefs if they had also seen the humorous videos. These findings suggest that people distinguish beliefs about vaccines, which deteriorated after exposure to the videos, from beliefs about vaccination, which improved. We recommend using humor when appropriate and focusing on the outcomes of vaccination, rather than on the vaccines themselves.

## 1. Introduction

India has borne a significant burden of the COVID-19 pandemic. By June 2022, India had reported more than 43 million cases, with an overall prevalence of 3.2% [1]. India’s mass vaccination campaign against COVID-19 was initially plagued by vaccine shortages and unexpected delays, but by April 2022, approximately 73% of the population had received their first dose, with 62% receiving both the first and second doses of the vaccine [2]. The Indian government provided the first two doses to the population for free. At the time of writing, booster doses are being offered only upon payment, and only 1.8% of the population have received one [2].

A peculiar phenomenon in India is the large gap (currently at 11%) between the proportion of people who have received both doses versus those who have received only the first dose. The gap is at its greatest among younger adults (aged 18–44), who are more likely to be partially vaccinated than older adults [3]. India recommends a relatively long 16-week wait between the first and second doses [4]. Despite this wait, it is difficult to explain this gap in vaccination, especially given that both doses are offered for free, but it does portend a challenge moving forward. In contrast, in many higher-income countries, the second dose was followed by a booster and, for vulnerable populations, a second booster dose has even been recommended and made available to the general public [5]. Given the challenges in India in administering the second dose more broadly, persuading the population to take on subsequent booster doses may be a particularly difficult task. It is thus important that we understand the underlying factors that inhibit vaccine uptake.

In India, the government has promoted vaccines through a variety of media, including government-produced public service announcements on television. These TV spots tend to adopt a more direct and factual approach in persuading people to get vaccinated by highlighting the dangers of not receiving the vaccines. Lacking a robust evaluation of such campaigns, we do not know how effective they have been in promoting vaccination drives in India, particularly among young adults.

The purpose of this paper is to test the efficacy of a video-based public service announcement that was proactively developed by considering three factors: the culture, the messenger, and the message itself. The overall research question we ask in this paper is: how do different aspects of the message such as the gender of the messenger, tone (humorous versus serious) and cultural orientation (collectivistic versus individualistic) affect beliefs about vaccines and vaccination?

### 1.1. Cultural Orientation: Collectivism and Individualism

Individualism and collectivism are sociocultural constructs that describe the degree of social cohesion in a society and an individual’s willingness to prioritize shared goals over personal goals [6,7]. In more individualistic-leaning societies, people are more likely to define themselves as unique individuals who value self-expression, while in collectivist-leaning societies, people are more likely to define themselves as members of a group [7,8]. Recognizing that no society is a monolith and that people fall along a spectrum, India is often characterized as being a more collectivist-leaning society [9]. Research suggests that in collectivist-leaning societies, individuals are more likely to act to protect the health of others, including through vaccination [10]. Emphasizing community benefits in health messaging has also been shown to increase vaccination in individualistic-leaning societies [11]. This phenomenon may function at both the community and individual levels; data comparing COVID-19 vaccine uptake across 50 countries found that vaccine intentions were higher in countries with more collectivistic cultures and that individuals from individualistic cultures who endorsed collectivistic values were more likely to accept a COVID-19 vaccine [12]. What is unknown, and what we investigate in this paper, is the extent to which the collectivistic or individualistic orientation in society can be made salient through vaccine videos and whether such heightened salience affects how people view and interpret the underlying message to get vaccinated.

### 1.2. Spokesperson Gender

Early communication research demonstrated that audiences are less likely to act on information that comes from “untrustworthy” sources [13]. In particular, trust in vaccine information has been strongly linked with trust in the source of that information [14]. In health communication, this body of research has shown that sociodemographic factors—including race/ethnicity, gender, and social status—can have a significant impact on the spokesperson’s credibility [15]. With COVID-19, a new wave of research has focused on the impact of different spokespersons when promoting vaccines to hesitant audiences [16]. Much of this research has focused on the United States and Europe, with fewer studies exploring messaging effects in lower- and middle-income countries.

Given the culturally specific nature of gender roles, it is unclear whether a male or female spokesperson would make the most credible source for vaccine information. Past research in Odisha, India (the site of the current work), focused on the impact of strong gender norms on health behaviors [17]. Despite India being a patriarchally oriented society, women do have a voice in family health decisions [18]. Women are also more likely to hold community health worker roles that are focused on promoting vaccines [19]. In this paper, we ask whether the same vaccine information coming from a female spokesperson is more or less persuasive in terms of getting vaccinated, in comparison to that coming from a male spokesperson. In conducting this analysis, we will also ask whether these spokesperson gender effects also vary by viewers’ genders.

### 1.3. Message Tone: Humorous or Serious

Vaccine promotion campaigns to date have tended to focus more on the presentation of factual information and less on the emotional and narrative elements of communication [20]. The use of humor has been largely overlooked [21]. Narrative persuasion theory suggests that using humor in a narrative style can influence viewers without triggering reactance [22]. Experimental research explicitly testing the use of humor in vaccination communication has been limited, but early studies are promising, suggesting that humorous messages may make audiences less reactive and more receptive to challenging information [23,24]. However, the same strategies can be used to disseminate misinformation about vaccines, particularly on social media platforms where humor and sarcasm may be easy ways to circumvent more restrictive content policies [25,26].

One challenge is that narrative messages—including humorous ones—leave more room for interpretation, which, in some cases, can lead viewers to unintended conclusions [22]. More recent research suggests that balancing “emotional flow” through health messaging may be an effective way to address health misinformation [21]. In this paper, we ask whether vaccination videos can induce humor and, if so, whether the heightened humor affects people’s reactions in a positive manner.

In this paper, we explore how factors related to cultural orientation, spokesperson gender, and tone (individually and jointly) affect vaccine beliefs.

## 2. Materials and Methods

### 2.1. Study Design

This paper reflects a collaborative effort between US-based research institutions and research and social media organizations in Odisha, India, to explore effective vaccine communication targeting young adults (ages 18–35). We conceptualized, wrote the scripts of, filmed, and produced eight Odia-language vaccine promotion videos (Appendix A). Videos were designed to be identical, with controlled experimental manipulations of three key variables: cultural orientation (collectivistic or individualistic), source gender (male or female protagonist), and tone (humorous or serious). These videos were then evaluated through a three-pronged approach, including an online experiment, qualitative research, and social network analysis of social media data [27]. This paper reports on only the online experiment.

Our online experiment followed a 2 (cultural orientation: collectivistic or individualistic) × 2 (spokesperson gender: male or female) × 2 (tone: humorous or serious) between-subjects design, with pre- and post-exposure measures. Ethical approval for this study was obtained from Institutional Review Boards of the two US-based institutions (Bloomberg School of Public Health at Johns Hopkins University and Loyola University of Maryland) and one from Odisha, India (D-Cor Consulting, SIGMA).

After expressing their consent to participate through an online consenting process, participants responded to questions on the Qualtrics platform, first asking them about their existing attitudes, beliefs, and behaviors pertaining to the COVID-19 vaccine. Subsequently, they were randomized to watch one of the eight videos, immediately after which they answered questions about their reactions to the video, vaccine-related attitudes and intentions, and their demographic and communication behaviors. The experiment was conducted over six days in February 2022. All questions and the videos themselves were in Odia. Questions were first written in English, translated into Odia, then backtranslated by a third party, and discrepancies were resolved through expert consultations.

### 2.2. Recruitment

Participants were recruited through the SwathsyaPlus Network (SPN) social media channels. SPN is a leading site for Odia-language health information, reaching more than six million viewers across India (and abroad) in different languages. SPN distributed the link to the survey by listing it on their website and through its WhatsApp channels of health workers and youth across the state. Upon completion, participants received a financial reward of Rs. 200 (USD 2.60), paid through Google Pay or similar methods. This amount was also deemed appropriate (striking a balance between serving as a sufficient incentive without being high enough to be coercive) by the Institutional Review Boards.

### 2.3. Participants

To be eligible, all participants had to be between the ages of 18 and 35, reside in India, own or have access to a cell phone, and be able to read and complete the survey in Odia. For this study, we defined young adults as those between 18 and 35, based on two factors: stratified case data from Odisha suggested the greatest burden of COVID falls among adults aged under 40, and input from our local collaborators suggested that 35 made a more logical generational cut-off point. Because participants had to provide their mobile phone numbers to get paid, we were able to exclude repeated participation by the same individual. The flowchart depicting enrollment procedures, inclusion and exclusion criteria, and missing values is shown in Figure 1.

### 2.4. Experimental Manipulations

We wrote a script describing the social interactions that occur at a young child’s birthday party attended by half a dozen family members. In the party, the child’s uncle or aunt (who also served as the main protagonist advocating for vaccines) interacts with the child’s mother in the presence of another family member who expresses skepticism about vaccines. Four videos featured a female as the main protagonist; the other four videos featured a male protagonist. Four of the eight videos were serious or matter-of-fact in tone; the other four were humorous in tone and dialog. Four of the eight videos discussed vaccines in the first-person singular, using phrases such as “I need to leave the party early so I can get vaccinated” and “the vaccine will protect me”. The other four used the first-person plural, with phrases such as “I need to leave the party early so that my family and I can get vaccinated” and “the vaccines will protect my family and me”. Apart from these three manipulations, the videos were otherwise identical. Indeed, to control for lighting, wind conditions, and other externalities, the videos were shot in parallel: for each segment of the scene, the camera angle was fixed, and eight different takes were recorded before moving on to the next scene.

Prior to shooting the videos, the scripts (translated into Odia) were reviewed by members of a community advisory board comprising a young married couple with children, a public health specialist, a community leader, a journalist, and community health workers. Feedback from the advisory board was used to further refine the scripts.

### 2.5. Measures

#### 2.5.1. Vaccine Attitudes and Vaccination Beliefs

The primary outcomes of interest were vaccine attitudes and beliefs about vaccination. Participants were asked to assess, on a five-point Likert scale ranging from “strongly disagree” to “strongly agree”, the extent to which COVID-19 vaccines were important, safe, effective, necessary, convenient, and affordable (adapted from Quinn et al., 2019) [28]. In addition, we assessed participants’ level of agreement with four other statements, related to the extent to which vaccinated people are likely to pass COVID-19 to others, that vaccination offers protection against severe illness or death, that vaccination protects one’s family, and that vaccination protects oneself. Responses to these nine questions were subjected to a principal component factor analysis with Varimax rotation, which yielded two factors, with eigen values of 6.5 and 1.1, respectively. These two factors explained 65.1% and 11.4% of the variance. The first five items related to vaccine attitudes and were averaged into an index (Cronbach’s α = 0.95). The latter four items related to beliefs about vaccination and were averaged into an index (Cronbach’s α = 0.87). Thus, while the first factor relates to people’s general attitudes about the COVID-19 vaccine, the second pertains to beliefs about getting vaccinated. The latter is particularly focused on vaccine efficacy. The same set of questions were also asked after exposure to the experimental video, and the reliability scores were comparable: Cronbach’s α = 0.89 for post-exposure vaccine attitudes and Cronbach’s α = 0.87 for post-exposure vaccination beliefs.

#### 2.5.2. Vaccination Status

We asked a series of binary questions, with participants indicating whether they had received first, second, and third doses of the COVID-19 vaccine. We coded this as a quantitative variable, ranging from 0 (no vaccines) to 3 (received the first booster dose).

#### 2.5.3. Demographic and Other Covariates

Covariates used in our models included demographic information about gender (male, female, other); age (calculated by asking participants’ year of birth); education (number of years of formal schooling); and residence (urban or rural setting).

#### 2.5.4. Manipulation Check Questions

To check the manipulation of message appeal, we asked participants three questions that asked whether the video they saw showed the benefits of vaccination, risks of not vaccinating, and the importance of vaccinating for individuals or for families or communities. One point was awarded for each correct answer (depending on whether the participant was in the individualistic or collectivistic condition).

To check the gender manipulation, participants were asked whether the person emphasizing the importance of vaccination was a man or a woman.

To check the humor manipulation, participants expressed their level of agreement or disagreement on a five-point scale as to the extent to which the video was humorous and the extent to which “it made me laugh”. Responses were averaged into an index (α = 0.91).

### 2.6. Statistical Analyses

We used chi-squares to test the extent to which demographic variables differed across the manipulated conditions (Table 1). To answer our primary research question, we conducted both bivariate and multivariate tests. For each outcome variable, we computed differences between pre-exposure and post-exposure values through paired *t*-tests and compared these differences (through independent sample *t*-tests) for each level of the independent variable (e.g., we compared changes in vaccine attitudes from pre-exposure to post-exposure in the humorous condition and compared it with the corresponding difference in the serious condition). In the multivariate models, through linear regressions, we first controlled for prior (i.e., pre-exposure) values, demographics, and vaccination status; we added the experimental manipulations (and the manipulation-check variables) and tested for main effects. Subsequently, we tested for interactions across the independent variables. All tests were conducted through SPSS version 25. For all analyses, a *p*-value of 0.05 was taken as the threshold for statistical significance. Main outcomes were also examined for normal distribution assumptions.

## 3. Results

### 3.1. Description of Sample

Table 1 shows the description of the sample in our study. Approximately 15% of the sample was between 18 and 21 years old, and another quarter of the sample was between 30 and 35 years old (our cut-off age), with a majority of our sample falling between 22 and 29 years old. This was also a well-educated sample: 38% were college graduates, with another 38% having attended college to some degree. Only 5% of the sample had up to a primary education. Approximately 57% reported living in a rural area. In terms of gender, our sample comprised 27% female participants. Most of the participants (69%) reported watching the video twice; another 24% reported watching the video more than twice. Most of our sample (77%) was fully vaccinated with both doses, and an additional 19% had also received a booster dose, bringing the fully vaccinated value to 95% of our sample.

Table 1 also shows how the demographic characteristics differed between each of the two arms of the three independent variables (individualistic or collectivistic orientation, humorous or serious tone, and male or female protagonist). Across all three independent variables, we saw no significant differences in demographic characteristics, in the frequency of watching the videos, or in vaccination status (none of the chi-squared tests were significant). The randomization process thus appears to have resulted in similar group characteristics in both arms of each independent variable.

### 3.2. Bivariate Analyses: Effects of Orientation, Tone, and Protagonist Gender

We tested the main effects of the three independent variables in two ways. First, we conducted bivariate tests, investigating whether the change in vaccine attitudes from pre-exposure to post-exposure and change in beliefs about vaccination was significant for each of the three independent variables. The results are shown in Table 2. Second, we conducted multivariate tests (results reported subsequently).

We observed that vaccine beliefs became more negative among those who saw both the individualistic (*M*_pre_ = 4.24, SD = 0.81; *M*_post_ = 4.19, *SD* = 0.73; *t* = 2.86, *p* < 0.01) and collectivistic (*M*_pre_ = 4.22, *SD* = 0.81; *M*_post_ = 4.15, *SD* = 0.73; *t* = 4.15, *p* < 0.001) conditions.

The serious video also resulted in more negative vaccine beliefs (*M*_pre_ = 4.26, *SD* = 0.79; *M*_post_ = 4.16, *SD* = 0.75; *t* = 5.63, *p* < 0.001), but the corresponding decline in the group exposed to the humorous video was not significant.

Those exposed to the male protagonist also developed more negative beliefs about vaccines (*M*_pre_ = 4.23, *SD* = 0.81; *M*_post_ = 4.17, *SD* = 0.75; *t* = 3.30, *p* < 0.01), as did those exposed to the female protagonist (*M*_pre_ = 4.23, *SD* = 0.81; *M*_post_ = 4.16, *SD* = 0.75; *t* = 3.72, *p* < 0.001).

Across each of the three experimental manipulations, a difference-in-difference model tested the decline in beliefs in one condition with that in the other. Neither the (collectivistic or individualistic) orientation manipulation (*t* = 0.84, *p* > 0.05) nor the protagonist gender manipulation (*t* = 0.16, *p* > 0.05) resulted in significant differences. The tone manipulation, however, resulted in significant differences (*t* = 3.01, *p* < 0.01), signifying that, while exposure resulted in greater skepticism overall, being exposed to the humorous video (as opposed to the serious video) resulted in a reduced rate of skepticism.

Across most of the experimental conditions, exposure to the video resulted in more positive vaccine attitudes. Exposure to the individualistic video resulted in more positive beliefs about vaccination (*M*_pre_ = 4.10, SD = 0.78; *M*_post_ = 4.14, *SD* = 0.76; *t* = 2.08, *p* < 0.05), as did exposure to the collectivistic video (*M*_pre_ = 4.08, *SD* = 0.81; *M*_post_ = 4.13, *SD* = 0.77; *t* = 2.87, *p* < 0.01).

The serious video also resulted in more positive attitudes about vaccination (*M*_pre_ = 4.08, SD = 0.78; *M*_post_ = 4.13, *SD* = 0.76; *t* = 2.71, *p* < 0.01), as did the humorous video (*M*_pre_ = 4.10, SD = 0.81; *M*_post_ = 4.14, *SD* = 0.76; *t* = 2.23, *p* < 0.05).

Exposure to the male protagonist resulted in more positive beliefs about vaccination (*M*_pre_ = 4.07, *SD* = 0.82; *M*_post_ = 4.14, *SD* = 0.75; *t* = 4.02, *p* < 0.001), but exposure to the female protagonist did not (*t* = 0.80, *p* > 0.05).

A difference-in-difference analysis showed that the improvement in beliefs about vaccination was affected by the manipulation of the protagonist’s gender (*t* = 2.38, *p* < 0.05), but not by the orientation or tone manipulations. Those who saw the videos with the male protagonist were significantly more likely to have improved their beliefs about vaccination, as compared to those who saw those with the female protagonist.

### 3.3. Bivariate Analyses: Effects of Orientation, Tone, and Protagonist Gender

We tested the main effects of the three independent variables in two ways. First, we conducted bivariate tests, investigating whether vaccine attitudes changed from pre-exposure to post-exposure and whether changes in beliefs about vaccination were significant for each of the three independent variables. The results are shown in Table 2. Second, we conducted multivariate tests (results reported subsequently).

We observed that vaccine attitudes became more negative among those who saw both the individualistic (*M*_pre_ = 4.24, *SD* = 0.81; *M*_post_ = 4.19, *SD* = 0.73; *t* = 2.86, *p* < 0.01) and collectivistic (*M*_pre_ = 4.22, *SD* = 0.81; *M*_post_ = 4.15, *SD* = 0.73; *t* = 4.15, *p* < 0.001) conditions.

The serious video also resulted in more negative vaccine attitudes (*M*_pre_ = 4.26, *SD* = 0.79; *M*_post_ = 4.16, *SD* = 0.75; *t* = 5.63, *p* < 0.001), but the corresponding decline in the group exposed to the humorous video was not significant.

Those exposed to the male protagonist also developed more negative vaccine attitudes (*M*_pre_ = 4.23, *SD* = 0.81; *M*_post_ = 4.17, *SD* = 0.75; *t* = 3.30, *p* < 0.01), as did those exposed to the female protagonist (*M*_pre_ = 4.23, *SD* = 0.81; *M*_post_ = 4.16, *SD* = 0.75; *t* = 3.72, *p* < 0.001).

Across each of the three experimental manipulations, a difference-in-difference model tested the decline in vaccine attitudes in one condition with that in the other. Neither the (collectivistic or individualistic) orientation manipulation (*t* = 0.84, *p* > 0.05) nor the protagonist gender manipulation (*t* = 0.16, *p* > 0.05) resulted in significant differences. The tone manipulation, however, resulted in significant differences (*t* = 3.01, *p* < 0.01), signifying that, while exposure resulted in greater skepticism overall, being exposed to the humorous video (as opposed to the serious video) resulted in reducing the rate of skepticism.

Across most of the experimental conditions, exposure to the video resulted in more positive beliefs about vaccination. Exposure to the individualistic video resulted in more positive beliefs about vaccination (*M*_pre_ = 4.10, *SD* = 0.78; *M*_post_ = 4.14, *SD* = 0.76; *t* = 2.08, *p* < 0.05), as did exposure to the collectivistic video (*M*_pre_ = 4.08, *SD* = 0.81; *M*_post_ = 4.13, *SD* = 0.77; *t* = 2.87, *p* < 0.01).

The serious video also resulted in more positive beliefs about vaccination (*M*_pre_ = 4.08, *SD* = 0.78; *M*_post_ = 4.13, *SD* = 0.76; *t* = 2.71, *p* < 0.01), as did the humorous video (*M*_pre_ = 4.10, *SD* = 0.81; *M*_post_ = 4.14, *SD* = 0.76; *t* = 2.23, *p* < 0.05).

Exposure to the male protagonist resulted in more positive beliefs about vaccination (*M*_pre_ = 4.07, *SD* = 0.82; *M*_post_ = 4.14, *SD* = 0.75; *t* = 4.02, *p* < 0.001), but exposure to the female protagonist did not (*t* = 0.80, *p* > 0.05).

A difference-in-difference analysis showed that the positive change in beliefs about vaccination was affected by the manipulation of the protagonist’s gender (*t* = 2.38, *p* < 0.05), but not by the orientation or tone manipulations. Those who saw the video with the male protagonist were significantly more likely to have more positive change in their beliefs about vaccination, as compared to those who saw the video with the female protagonist.

### 3.4. Multivariate Analyses: Effects of Orientation, Tone, and Protagonist Gender

The results of our multivariate analyses are shown in Table 3. In the first set of analyses, with post-exposure vaccine attitudes as the dependent variable, we included as control variables vaccination status, pre-exposure beliefs, demographics, and manipulation check outcomes. The three experimental manipulations were then added as independent variables, followed by interactions among the independent variables. The same steps were followed for analyzing post-exposure beliefs about vaccination as the outcome.

Vaccination status was not associated with vaccine attitudes, but pre-exposure beliefs were strongly associated with post-exposure beliefs (β = 0.67, *p* < 0.001). Age (β = 0.04 *p* < 0.01), education (β = 0.07, *p* < 0.001) and urban residence (β = 0.03, *p* < 0.05) were positively associated with vaccine attitudes. Collectivism was not associated with vaccine attitudes (which was true for both the manipulated variable as well as the manipulation check variable). The humorous video resulted in positive vaccine attitudes through both the manipulation (β = 0.03, *p* < 0.05) and the manipulation check (β = 0.07, *p* < 0.001). Although the gender manipulation was not associated with vaccine attitudes, those who correctly identified the protagonist’s gender had more positive attitudes (β = 0.07, *p* < 0.001). A significant interaction effect was also observed between the gender manipulation check variable and exposure to the humor video. The pattern of the interaction is shown in Figure 2, illustrating that while those who correctly identified the protagonist gender had more positive attitudes about vaccines, those who did not correctly identify the protagonist’s gender were more likely to harbor positive attitudes if they were exposed to the humor (as opposed to the serious) video. Overall, the model explained 50.6% of the variance in vaccine attitudes.

Prior vaccination status was not associated with beliefs about vaccination, but pre-exposure belief was a significant predictor of post-exposure belief (β = 0.69, *p* < 0.001). Age (β = 0.06, *p* < 0.001) and education (β = 0.06, *p* < 0.001) were positively associated with beliefs about vaccination. Among the manipulation check variables, only the ability to identify the protagonist’s gender was positively associated with more positive beliefs about vaccination (β = 0.07, *p* < 0.001). None of the three manipulations predicted vaccination beliefs. However, a significant interaction effect was observed between the gender manipulation check and exposure to the humor video (β = −0.18, *p* < 0.001), and the pattern of the interaction is shown in Figure 3. As in the previous case, the ability to correctly identify the protagonist gender resulted in more positive beliefs about vaccination. Among those less able to identify the protagonist’s gender, those who saw the humor video had more positive beliefs in vaccination than those who saw the serious video.

## 4. Discussion

The primary purpose of this paper was to determine the extent to which three characteristics of vaccine communication messages affect people’s vaccine attitudes and beliefs about vaccination: the collectivistic or individualistic orientation of the message, gender of the vaccine proponent, and the humorous or serious tone of the video. Because the study was designed to answer the questions in a causal manner, we randomized the experimental conditions, which also required the independent variables to be successfully manipulated.

Our manipulation checks revealed that each of the variables was manipulated as intended. Participants in the collectivistic arm viewed the videos as highlighting families and communities more than individuals, as compared to those in the individualistic arm. Similarly, gender and humor were also manipulated successfully. The underlying process through which the videos were shot ensured consistency across the videos. This required shooting each scene eight different times in accordance with the manipulations, while still maintaining the consistency of the camera angle, sound, and other externalities.

The successful manipulations, however, did not result in corresponding differences in outcomes for two of the three predictors. Our data revealed that neither the collectivistic or individualistic orientation of the message nor the protagonist’s gender differentially affected beliefs about vaccines or about vaccination. One explanation for this finding may well be that the essence of the message—which highlighted the discussion between two adults, one vaccine-hesitant and the other vaccine-advocating—overshadowed the underlying differences in cultural orientation between whether individuals or larger groups would benefit from vaccination. In this rather charged debate, perhaps gender also ceased to rise to the same level of prominence as the debate itself. This, of course, is speculative and remains to be understood more definitively.

Of the three manipulations, only humor had a differential impact: those who viewed the humorous version of the videos, as compared to the serious ones, were less likely to develop negative attitudes about COVID-19 vaccines. The overall trend in our pre-post comparison was the deterioration of vaccine attitudes across all experimental conditions. This decline, however, was the lowest among those who viewed the humorous video. We suspect that the humor in the video led people to treat the entire message more lightly, with less vested interest in one or the other side of the vaccine debate—leading to reduced skepticism. This, too, remains to be explored more rigorously in future research.

This brings us to another significant finding: we found a notable difference in the outcomes we observed between vaccine attitudes, which became more negative post-exposure, and beliefs about vaccination, which became more positive. Watching any video, it appears, made audience members view vaccines in a more skeptical light, whereas their views about vaccination became more positive. In this distinction, vaccine-related attitudes pertain to how people evaluated the vaccines themselves (e.g., whether they were accessible, effective, etc.), whereas beliefs about vaccination pertain to how people evaluated the outcomes vaccination—including the efficacy of the vaccines. The primary implication from this finding seems to be that, for effective campaigns, we are better off focusing on the outcomes brought about by vaccination rather than on the vaccines themselves.

We are unable to tell definitively why people’s vaccine attitudes became more skeptical, but one likely reason pertains to how our videos ended: with considerable ambiguity about what exactly needed to be done. A topic of significant internal debate in our team centered around how extensively the videos should expound on only the positive aspects of vaccines, as opposed to inviting viewers to fill in the gap when someone on the video expresses some skepticism. In the end, we decided that viewers were already receiving a more direct appeal for vaccination through government outreach efforts and other means and that we could adopt a somewhat more realistic approach and invite greater audience participation by highlighting and then refuting vaccine skepticism, while leaving the final resolution up to the viewer. One noted challenge in conveying health messaging through humor is that a narrative style leaves more up to interpretation [22].

When focusing on vaccine content, it may be that, by giving airtime to *any* kind of skepticism, we endowed it with more credence or legitimacy. Indeed, many vaccine messaging studies have observed a “backfire effect”, where debunking claims inadvertently reinforces its resonance [29]. Another possibility is that when negative attitudes toward vaccines are depicted, people come to believe in the size of this imaginary crowd—that if such opinion is being displayed by one person, many others must also harbor such beliefs. This underlying idea of social norms, particularly around perceived harm, has a significant impact on vaccination behaviors but remains understudied in the context of COVID-19 vaccines [30,31]. While both pathways are possible explanations for the greater negative attitudes, they (and other possible explanations) remain to be tested more rigorously. What this experience does tell us is that messages must end with clarity, particularly when more than one opinion is being depicted.

We also found that humor can reduce negative evaluations of vaccine videos. When people can laugh with the message, they are less likely to harbor negative attitudes about what transpired. The use of humor, of course, must be tasteful and culturally sensitive, particularly when dealing with a topic, such as the COVID-19 pandemic, which has resulted in a lot of destruction. Indeed, this was one of the precautionary reactions from our community advisory board members when they first read the video scripts—that we needed to treat the topics with sensitivity. Hence, humor that was depicted in the videos focused more on the video production aspect, including incorporating humor in sound and editing, and not on the consequences of COVID-19. This insight aligns with Buijzen and Valkenburg’s typology of humor and underscores the need to align humor type to the target audience [32].

Finally, we note the interaction between humor and ability to identify protagonist gender, as depicted in Figure 2 and Figure 3. After watching the video, we asked participants whether the main vaccine proponent in the video was a man or woman, and we suspect that the answer is a proxy for how much attention participants were paying to the video. Those who correctly identified the vaccine proponent’s gender were likely paying more attention than those unable to do so (this latter group constituted about a quarter of the sample). Both Figure 2 and Figure 3 show, in essence, that inattention-driven attenuation in outcomes can be ameliorated a bit by humor: funny content can improve attention and thus improve outcomes. This finding aligns with a growing body of health and science communication research that points to the value of humor to gain and hold attention on social media [33].

### Strengths and Limitations

One strength of this study is its randomized design, which allows us to make causal attributions to the changes in viewers’ reactions after watching the video, which speaks to the internal validity. The real-world conditions in which the videos were viewed (as opposed to viewing them in a laboratory setting), coupled with depictions of a slice-of-life scenario, are aspects of the study that speak to external validity. In combination, these two factors add significant strengths to the study. We note, however, that our recruitment methods resulted in a significant compromise in the representativeness of our sample: we had mostly (73%) male participants, most of our sample had received both doses of the vaccine, and our sample was also well educated, with approximately 38% of the sample being college graduates (whereas the corresponding state average is less than ten percent). This is likely the product of our recruitment methods, which solicited participants from the SwasthyaPlus Network, coupled with publicity through existing WhatsApp user groups, both of which tapped into an online audience that is more educated and more likely to be male, as compared to the rest of the population. Given the source of recruitment, we also likely tapped into an audience that was already health conscious. Hence, generalizing to other populations must be done with caution.

Two other strengths of the underlying work were the high production values achieved by employing a professional filmmaking crew and the engaging video content, which was the result of scripts being developed through an extensive consultative process that included behavior change experts, scriptwriters, and local community stakeholders. We realize that the ability to manipulate content, tone, and gender in this way may be a luxury unavailable to most health professionals, as it requires significant resources. During pilot testing, we learned, however, that simple audio recordings that serve as proxies of the different manipulations can provide important information about how the content comes across for the viewers.

## 5. Conclusions

Among the questions addressed in this research is one regarding the aspects of the videos (tone, cultural orientation, or protagonist’s gender) that viewers are paying attention to. Because we created these videos to be disseminated on social media platforms, attention, retention, or even divided attention become key outcomes to be considered. For instance, we reported an outcome that some viewers focused on content that increased their skepticism, and we were able to distinguish differences between post-viewing beliefs regarding vaccines and vaccination.

Both Facebook and YouTube (where these videos are intended to appear) operate within the broader digital media ecosystem, and as such the messages contained in these videos must contend with competing messages, including mis- or disinformation about COVID-19 vaccines. There is also competition for attention at the level of message attributes within each video explored in this research. These messages, therefore, compete in what might be characterized as the attention economy in which the means of production (video) is impacted by the means of distribution (channel). Attention is conflated by the tone and appeal of our messaging and the protagonist’s gender, but all COVID-19 vaccine messaging in the digital realm is made more complex when considering the means of delivering the messaging (e.g., social media platforms where these videos will ultimately appear).

If gaining and holding attention is fundamental to successful health communication, then as traditional approaches or structures of power and authority regarding vaccine message delivery are challenged, attention becomes a vital resource to be considered. This study has addressed the issue of attention by analyzing differences between viewers’ perceptions of vaccines versus what they think about vaccination, noting what draws them in (humor) and what, for example, might increase vaccine skepticism. Moreover, the messages in all eight videos are delivered within a “slice-of-life” context, which distinguishes our approach from that of more traditional approaches, as it relies on ordinary people having routine conversations.

Given that attracting and retaining attention is a key resource in the attention economy, it is incumbent upon health communicators, perhaps more than ever, to utilize tactics such as those attempted in this study to bring greater focus to the most relevant aspects of the issue at hand. Operating within the ecology of the digital media system in which we must compete for attention, this study offers an approach to increase the likelihood that the right message will get through to the correct audience through the most appropriate medium. We invite future scholars to add to this work.

## Figures and Tables

**Figure 1 vaccines-10-01110-f001:**
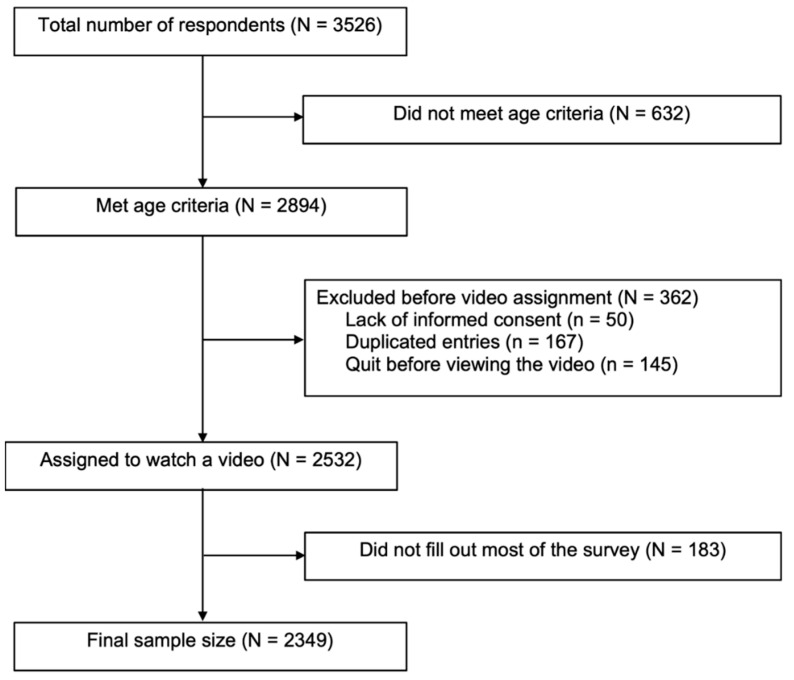
Study flowchart.

**Figure 2 vaccines-10-01110-f002:**
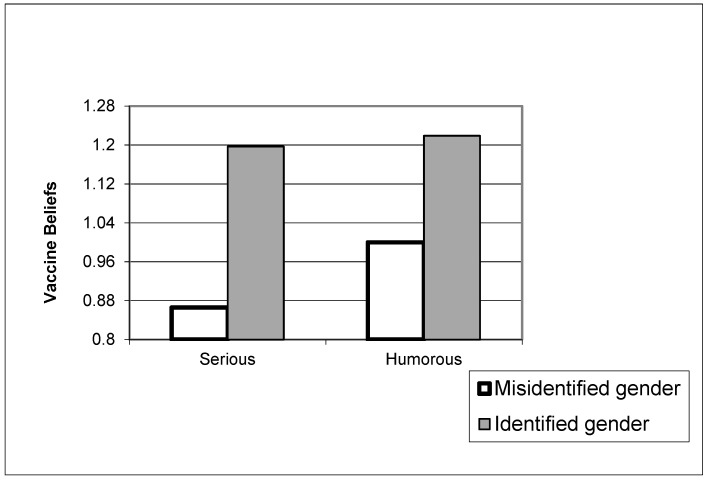
Vaccine attitudes as a function of humor and ability to identify protagonist’s gender.

**Figure 3 vaccines-10-01110-f003:**
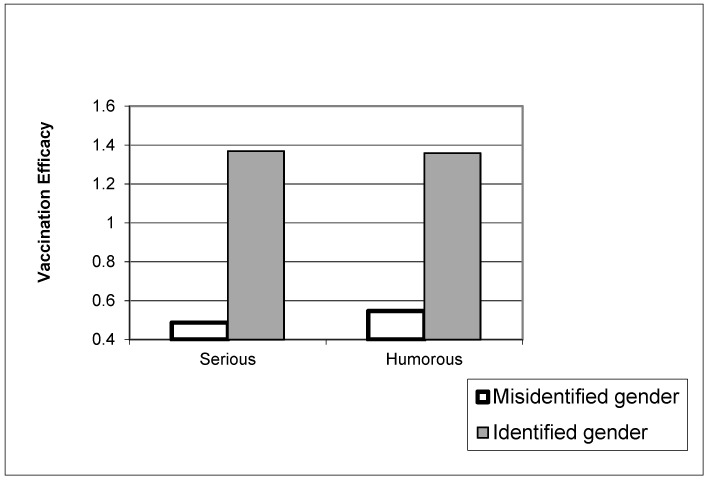
Vaccination beliefs as a function of humor and ability to identify protagonist’s gender.

**Table 1 vaccines-10-01110-t001:** Characteristics of the sample across experimental conditions.

			Manipulated Independent Variables								
			Orientation			Tone			Protagonist Gender		
	*N*	%	Indiv.	Coll.	*X* * ^2^ *	Ser.	Hum.	*X* * ^2^ *	Male	Fem.	*X* * ^2^ *
Age:											
<21 years	357	15.2	47.6	52.4		50.1	49.9		51.0	49.0	
21–39 years	649	27.6	48.1	51.9		47.1	52.9		50.7	49.3	
25–29 years	728	31.0	50.4	49.6		49.6	50.4		46.8	53.2	
≥30 years	615	26.2	50.6	49.4	0.67	51.7	48.3	0.44	54.1	45.9	0.07
Education:											
Up to primary	120	5.1	50.0	50.0		42.5	57.5		51.7	48.3	
Up to secondary	444	18.9	45.0	55.0		50.0	50.0		54.5	45.5	
Some college	898	38.2	48.7	51.3		50.4	49.6		49.4	50.6	
College graduate	886	37.7	52.3	47.7	0.09	49.3	50.7	0.44	49.2	50.8	0.27
Rural resident	1347	57.3	49.4	50.6	0.97	49.5	50.5	0.93	51.1	48.9	0.37
Female respondent	625	26.6	50.7	49.3	0.46	50.2	49.8	0.70	49.0	51.0	0.35
Viewing frequency:											
Once	157	6.7	52.2	47.8		53.5	46.5		48.4	51.6	
Twice	1619	68.9	49.2	50.8		48.7	51.3		50.2	49.8	
Three or more	573	24.4	49.2	50.8	0.76	50.8	49.2	0.41	51.7	48.3	0.73
Vaccination status											
None	9	0.4	33.3	66.7		55.6	44.4		44.4	55.6	
First dose only	94	4.0	56.5	43.6		48.9	51.1		56.4	43.6	
Fully vaccinated	1812	77.1	48.4	51.2		49.8	50.2		50.3	49.7	
Booster dosed	434	18.5	50.5	49.5	0.36	48.6	51.4	0.95	49.8	50.2	0.67

Notes: Indiv. = individualistic orientation. Coll. = collectivistic orientation. Ser. = serious tone. Hum. = humorous tone. Fem. = female protagonist. Ratios shown under the Orientation, Tone, and Protagonist Gender columns pertain to row percentages. None of the chi-squared values were significant, indicating that the randomization resulted in no underlying demographic differences between the two arms of the three independent variables (orientation, tone, and protagonist gender). For readability, we have not added the degrees of freedom for the chi-square tests; they are *k*-1 (where *k* is the number of levels for each variable; for example, *age has four levels and three degrees of freedom*).

**Table 2 vaccines-10-01110-t002:** Vaccine beliefs and vaccination beliefs in the three experimental conditions (independent variables).

	Vaccine Beliefs						Vaccination Efficacy Perceptions					
	Pre-Exposure		Post-Exposure				Pre-Exposure		Post-Exposure			
	*M*	*SD*	*M*	*SD*	*t_pre-post_*	t_DID_	*M*	*SD*	*M*	*SD*	*t_pre-post_*	t_DID_
Orientation:												
Collective	4.24	0.81	4.19	0.73	2.86 **		4.10	0.78	4.14	0.76	2.08 *	
Individual	4.22	0.81	4.15	0.77	4.15 ***	0.84	4.08	0.81	4.13	0.77	2.87 **	0.51
Tone:												
Serious	4.26	0.79	4.16	0.75	5.63 ***		4.08	0.78	4.13	0.76	2.71 **	
Humorous	4.20	0.083	4.17	0.75	1.41	3.01 **	4.10	0.81	4.14	0.76	2.23 *	0.34
Protagonist Gender:												
Male	4.23	0.81	4.17	0.75	3.30 **		4.07	0.82	4.14	0.75	4.02 ***	
Female	4.23	0.81	4.16	0.75	3.72 ***	0.16	4.12	0.77	4.13	0.77	0.80	2.38 *

Notes: “*t_pre-post_*” refers to the *t*-test that compares differences between pre-exposure and post-exposure values for each level of the independent variable. For example, 2.86 is the *t*-value that compares vaccine beliefs at the individual level, between pre-exposure and post-exposure. “*t_DID_*” refers to the *t*-test that compares differences: the gain in outcome at one level of the independent variable, compared to the gain in outcome at the other level of the independent variable. For example, 0.84 is the t-statistic comparing the pre-post difference at the individual level with the pre-post difference at the collective level. Vaccine beliefs, scored on five-point scales, represent beliefs about access, safety, affordability, convenience, and effectiveness of vaccines; higher numbers represent more positive beliefs. Vaccination beliefs pertain to the effects of getting vaccinated; higher numbers represent more positive perceptions. * *p* < 0.05, ** *p* < 0.01, *** *p* < 0.001. *t-test degrees of freedom ranged from 1159 to 1188; for readability of the table, they are not shown*.

**Table 3 vaccines-10-01110-t003:** Vaccine attitudes and vaccination beliefs as a function of orientation, tone, and protagonist gender from multiple regression equations.

	Vaccine Attitudes		Vaccination Beliefs	
	b	R^2^ (%)	b	R^2^ (%)
Full vaccination status	0.00		−0.03	
Pre-exposure beliefs (or efficacy)	0.67 ***		0.69 ***	
Demographics				
Female	−0.02		0.00	
Age	0.04 **		0.06 ***	
Education	0.07 ***		0.06 ***	
Rural residence	−0.03 *		−0.01	
Manipulation check variables				
Collectivism	−0.01		0.02	
Humor	0.07 ***		0.01	
Gender ^a^	0.07 ***		0.07 ***	
Independent variables				
Collectivism	−0.02		0.00	
Humor	0.03 *		0.00	
Gender (Female = 1, Male = 0)	0.00	0.500 ***	−0.02	
Interaction variable			−0.18 ***	0.512 ***
Manipulation check gender x humor	−0.24 ***	0.506***		0.515 ***

Notes: ^a^ Refers to the accuracy with which participants identified the protagonist gender (0 = inaccurate, 1 = accurate). Cell entries are standardized betas from regression equations with all the main effects (but without the interactions) entered simultaneously. The interaction term was entered in the second step. * *p* < 0.05, ** *p* < 0.01, *** *p* < 0.001.

## Data Availability

The survey instrument and the data presented in this study are available on request from the corresponding author. The data are not publicly available due to privacy concerns.

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
