# Peer review of "Promoting Vaccination in India through Videos: The Role of Humor, Collectivistic Appeal and Gender"

_vaccines, 2022, doi:10.3390/vaccines10071110_

Round 1

Reviewer 1 Report

After careful consideration, I fell that the manuscript entitled “Promoting Vaccination in India through Videos: The Role of Humor, Collectivistic Appeal, and Gender” has merit, but is not suitable for publication as it currently stands. Therefore, my decision is "Major Revision."

There are some aspects that need review:

a)    Section 2.5.1: It is necessary to describe what alpha is. Is it Cronbach's alpha? Are the observed values adequate? How was the likert scale coded to form the two scores? What is the range of the score values?

b) Section 2.5.2: to replace the term “continuous variable” for “quantitative variable”.

c) Section 2.6: The study suggests a 2x2x2 between-subjects design. However, comparisons were performed using t-tests.Why did the authors not perform the comparison using a three-way ANOVA when comparing the three effects (type of appeal, message tone, promoter gender)?

In addition, several important pieces of information were not described in this section:
- the chi-squared test;
- further specify the t test: in which situations a paired test was performed and in which an unpaired test was performed (Table 2);
- the program used to perform the analyses;
- the adopted significance level of significance;
- the verification of the normality assumption.

d) Table 1 is unformatted and hard to read. Furthermore, since the authors chose to present the chi-square value instead of the p-value, it is also important to present the degrees of freedom.

e) Table 2 is unformatted and hard to read. Furthermore, since the authors chose to present the t value instead of the p-value, it is also important to present the degrees of freedom.

f) Table 3 is unformatted. In addition, the authors could present the confidence intervals of beta coefficients

Author Response

Thank you for your thoughtful reviews. 

Reviewer 2 Report

Jamison et al. describe a study where 8 videos were deployed amongst young adults from a specific geographical location to evaluate the influence of collectivist cultural messages, humor, gender of the protagonist, etc. on an individual's perception about vaccines and vaccination. The study is interesting and arguably equally valid for all vaccines at all times (not just during the Covid pandemic). However, I found the choice of the 3 factors very peculiar and I wondered why other relevant factors such as the content, the social setting of the script used, the citations used (scientific facts versus hearsay) weren't also evaluated. I would rate the article as an interesting read but wouldn't give too much weight to the relevance of the conclusions in a global setting. The language used is confusing in some places. I am including some comments below that would greatly improve the comprehension and reach of the article:

Line 36-37: Numbers to be updated to most recent (June 2022)

Line 38: Replace 'late' with the rough timing or with a comparator (later than in the Western world, x months after the start of the pandemic) for clarity

Line 39 and 43 and other places: Use symbol for percent (%)

Line 45: Remove 'unusually' or replace with 'relatively'. For the vaccines offered (Covishield and Covaxin, a 4-month gap is not unusual and is in line with the manufacturer's guidelines).

Lines 53-54: Rephrase. 'Acquiring a vaccine dose' is not the correct meaning

Line 59: Expand 'young adults (the focus of our paper) and specify the age group targeted and criteria used to filter candidates on the basis of their age. Or insert reference to section 2.3.

Line 68: Replace 'common' with 'community' (or similar) for clarity and accuracy

Line 125: rephrase, check grammar

Lines 158-159: specify the names of the 3 institutions mentioned

Figure 1: Replace 'quit before seen the video' with 'quit before seeing the video'

Figure 1: Define 'most' in 'did not fill out most of the survey' in terms of >50% or >20% or whatever accurate

Line 193: hyphenate 'matter-of-fact'

Tables 1 and 2 and other places: fix spacing for clarity and use a 0 in front of numbers such as .36, .46, .97 to avoid misinterpretation

Lines 350-352: Meaning not clear. Please rephrase.

Lines 361-362: Explain 'improved beliefs' or replace with 'positive outlook' for clarity.

Lines 515-517: Replace 'skeptical light' with 'skepticism' and adjust grammar

Overall: Please proof-read for language, spelling and missing punctuation.

Supplementary information: Include links to the videos (youtube versions) and/or the questionnaire used to add to the credibility of the article

Author Response

Thank you for your thoughtful reviews. Please see attached. 

Round 2

Reviewer 1 Report

I am satisfied with the corrections presented by the authors.

Author Response

Thank you for your thoughtful reviews.